# A New Adaptive Remote Sensing Extraction Algorithm for Complex Muddy Coast Waterline

Ziheng Yang [1,2], Lihua Wang [1,3,4,*], Weiwei Sun [1,3], Weixin Xu [2], Bo Tian [4], Yunxuan Zhou [4], Gang Yang [1,3] and Chao Chen [5]

1   Department of Geography and Spatial Information Techniques, Ningbo University, Ningbo 315211, China; 2015045020@cuit.edu.cn (Z.Y.); sunweiwei@nbu.edu.cn (W.S.); yanggang@nbu.edu.cn (G.Y.)
2   School of Resources and Environment, Chengdu University of Information Technology, Chengdu 610225, China; weixin.xu@cuit.edu.cn
3   Institute of East China Sea, Ningbo University, Ningbo 315211, China
4   State Key Laboratory of Estuarine and Coastal Research, East China Normal University, Shanghai 200062, China; btian@sklec.ecnu.edu.cn (B.T.); zhouyx@sklec.ecnu.edu.cn (Y.Z.)
5   Marine Science and Technology College, Zhejiang Ocean University, Zhoushan 316022, China; chenchao@zjou.edu.cn
*   Correspondence: wanglihua1@nbu.edu.cn

**Abstract:** Coastline is an important geographical element of the boundary between ocean and land. Due to the impact of the ocean-land interactions at multiple temporal-spatial scales and the intensified human activities, the waterline of muddy coast is undergoing long-term and continuous dynamic changes. Using traditional remote sensing-based waterline extraction methods, it is difficult to achieve ideal results for muddy coast waterlines, which are faced with problems such as limited algorithm stability, weak algorithm migration, and discontinuous coastlines extraction results. In response to the above challenges, three different types of muddy coasts, Yancheng, Jiuduansha and Xiangshan were selected as the study areas. Based on the Sentinel-2 MSI images, we proposed an adaptive remote sensing extraction algorithm framework for the complex muddy coast waterline, named AEMCW (Adaptive Extraction for Muddy Coast Waterline), including main procedures of high-pass filtering, histogram statistics and adaptive threshold determination, which has the capability to obtain continuous and high-precision muddy coastal waterline. NDWI (Normalized Difference Water Index), MNDWI (Modified Normalized Difference Water Index) and ED (Edge Detection) methods were selected to compare the extraction effect of AEMCW method. The length and spatial accuracy of these four methods were evaluated with the same criteria. The accuracy evaluation presented that the length errors of ED method in all three study areas were minimum, but the waterline results were offset more to the land side, due to spectral similarity, turbid water and tidal flats having similar values of NDWI and MNDWI. Therefore, the length and spatial accuracies of NDWI and MNDWI methods were lower than AEMCW method. The length errors of the AEMCW algorithm in Yancheng, Jiuduansha, and Xiangshan were 14.4%, 18.0%, and 7.7%, respectively. The producer accuracies were 94.3%, 109.6%, and 94.2%, respectively. The user accuracies were 82.4%, 92.9%, and 87.5%, respectively. These results indicated that the proposed AEMCW algorithm can effectively restrain the influence of spectral noise from various land cover types and ensure the continuity of waterline extraction results. The adaptive threshold determination equation reduced the influence of human factors on threshold selection. The further application on ZY-1 02D hyperspectral images in the Yancheng area verified the proposed algorithm is transferable and has good stability.

**Keywords:** adaptive remote sensing extraction algorithm; muddy coast; waterline; high-pass filtering; Sentinel-2

## 1. Introduction

The coastal zone is the ocean–land interaction region with important ecological functions and great resource values, which is the forefront for the development of the marine

economy, and plays an important role in the national economic sustainable development strategy [1,2]. Coastline is an important geographical element for the boundary between land and ocean. Strengthening coastline monitoring cannot only help us to understand the characteristics and rules of coastal environmental and ecological processes scientifically, but it can also support the making of scientific and reasonable integrated decisions between land and ocean. Accurate and promptly coastline monitoring is also helpful to promote the sustainable development, protection and management of resources, environment and ecosystem in coastal zone [3].

Compared with other natural coast such as bedrock coast, sandy coast and biogenic coast, muddy coast has more prominent dynamic features as it is influenced by land and ocean interaction at multiple temporal-spatial scales and human activities. Depending on the geomorphic types, the muddy coast can be divided into three types: plain muddy coast, delta muddy coast and harbor muddy coast. The plain muddy coast and delta muddy coast are closely related to the rivers entering the ocean. When the river supplies enough sediments to the ocean, the sediments which spread along the coast and ocean promote the seaward siltation of the muddy coast [4]. Otherwise, muddy coast retreats back to land when the rivers cannot supply sufficient sediments. Harbor muddy coasts are usually developed in harbors that are located at coastal hilly areas. Due to the tortuous coast and narrow harbor, harbor muddy coast is generally small in size. The silts in the bay mainly come from the sediments carried by the inland rivers and the resuspension of bottom ocean sediments. Therefore, the location of the muddy coastline is highly dynamic. This not only includes long-term changes caused by sediment transport in upstream rivers, but also short-term changes caused by tides and other influences. The observation period of traditional coastal survey is too long to meet the needs of real-time monitoring or decision-making assistance.

In recent years, remote sensing technologies such as satellite and UAV (unmanned aerial vehicle) have become important tools for coastline extraction because of the advantages of convenient data acquisition, large-scale monitoring and high temporal-spatial resolution [5]. In fact, the ocean-land boundary based on remote sensing image is instantaneous waterline. The coastline can be obtained by combining the instantaneous waterline with the tide level information. Therefore, the accurate extraction of the waterline based on remote sensing image is an important basis for coastline extraction [6]. The common methods of waterline extraction based on remote sensing can be divided into two categories, VI (visual interpretation) and automatic extraction [7]. The VI method has the advantages of simple operation and high extraction accuracy, but also has the disadvantages of heavy workload and low efficiency. Compared to the VI method, the automatic extraction method has higher extraction efficiency. Generally, it can be divided into index analysis-based method [8–12], edge detection-based method [7,13–15], threshold segmentation-based method [16–20], region growth-based method [7], neural network-based method [21–24] and sub-pixel method [16,25–29].

The index analysis-based method distinguishes ocean and land by calculating water indexes such as NDWI (Normalized Difference Water Index) and MNDWI (Modified Normalized Difference Water Index) [30]. The physical meaning of these methods is clear, which takes advantage of the fact that the reflectance of water is near to zero in the near-infrared band and high in the green band [31]; yet, the thresholds of these methods are unstable [10–12]. As for the different remote sensing sensor, image acquisition time and coverage area, the thresholds need to be re-determined. In the ocean-land junction area of muddy coast, due to the high water content of tidal flats and high suspended sediments concentration of offshore water, the NDWI and MNDWI values of land and water have little differences, which causes the waterline blurred. The ED (Edge Detection) method applies an edge detection operator to detect the position of step changes in image gray values [7,14,15], it is simple and effective. But the edge information of the ocean-land conjunction area in muddy coast is too weak to make the extracted waterline continuous [32]. The threshold segmentation-based method is suitable for images with strong contrast

between target and background, but it is easily affected by land cover types with similar spectral characteristics [18,20]. Besides, the unified global threshold cannot guarantee the accuracy of the local area while ensuring the overall accuracy. Adaptive threshold determination can reduce the influence of subjective factors and make the threshold more accurate. Chan et al. proposed a new adaptive thresholding method using variational theory. The method requires only one parameter to be selected and the adaptive threshold surface can be found automatically from the original image [33]. Wei et al. proposed an adaptive classification algorithm of water LiDAR point clouds to distinguish the water points from the land points in complex landscapes [34]. This algorithm can also achieve a classification accuracy higher than 99% in complex landscapes with mudflats and inland plains. More and more researchers are applying adaptive methods to their studies. The basic idea of the region growth-based method is using similar pixels to form a complete region. This method is applicated widely and can obtain continuous waterline results. However, it is difficult to choose a suitable growth rule, which can be easily affected by noise and is not suitable for areas with various land cover types [35]. The principle of the neural network-based method simulates the structure and function of the human brain neuron network [36]. The human-like thinking can be achieved by establishing a simple model to form different networks according to different linking methods [22]. The recent decades have seen the driving advance of neural networks in various visual recognition fields such as object detection [37,38], image classification [39], and semantic segmentation [40,41]. However, the calculation process of the neural network method is complicated and it needs pixel samples as pure as possible [23,24]. The sub-pixel method can obtain waterline results better than pixel level by reducing the influence of mixed pixels on waterline extraction. However, it is more difficult to unmix the mixed pixels [26,27]. Therefore, using the above traditional remote sensing-based waterline extraction methods, it is difficult to achieve ideal results for muddy coast waterline, which is faced with problems such as limited algorithm stability, weak algorithm migration, and discontinuous coastlines extraction results.

Aiming at solving these problems, we proposed a new adaptive remote sensing extraction algorithm framework for complex muddy coast waterline, named AEMCW (Adaptive Extraction for Muddy Coast Waterline). This algorithm combined high-pass filtering, histogram statistics and morphological processing. Unlike the traditional ED method that uses edge detection operator to directly extract edge information, AEMCW method firstly extracts the low-frequency information and performs morphological processing on it. Then, the edge information is represented by extracting the boundary of low frequency information. This improvement can make the extracted waterline complete and continuous. Meanwhile, the band used for waterline extraction in AEMCW method is less affected by spectral similarity. The AEMCW method proposed in this paper has better adaptability and accuracy than traditional waterline extraction method in muddy coast. The structure of this paper is as follows: The second section introduces the study area and remote sensing data, and elaborates the AEMCW algorithm; the third section shows the results of muddy coast waterline extraction; the fourth section discusses the accuracy of the waterline extraction and the comparison between our results and the waterlines extracted from traditional methods; the fifth section is the conclusion.

## 2. Materials and Methods

### 2.1. Study Area

Three typical study areas were selected based on the landform types of the muddy coast, which were the Yancheng plain muddy coast, the Jiuduansha delta muddy coast and the Xiangshan harbor muddy coast (Figure 1), respectively.

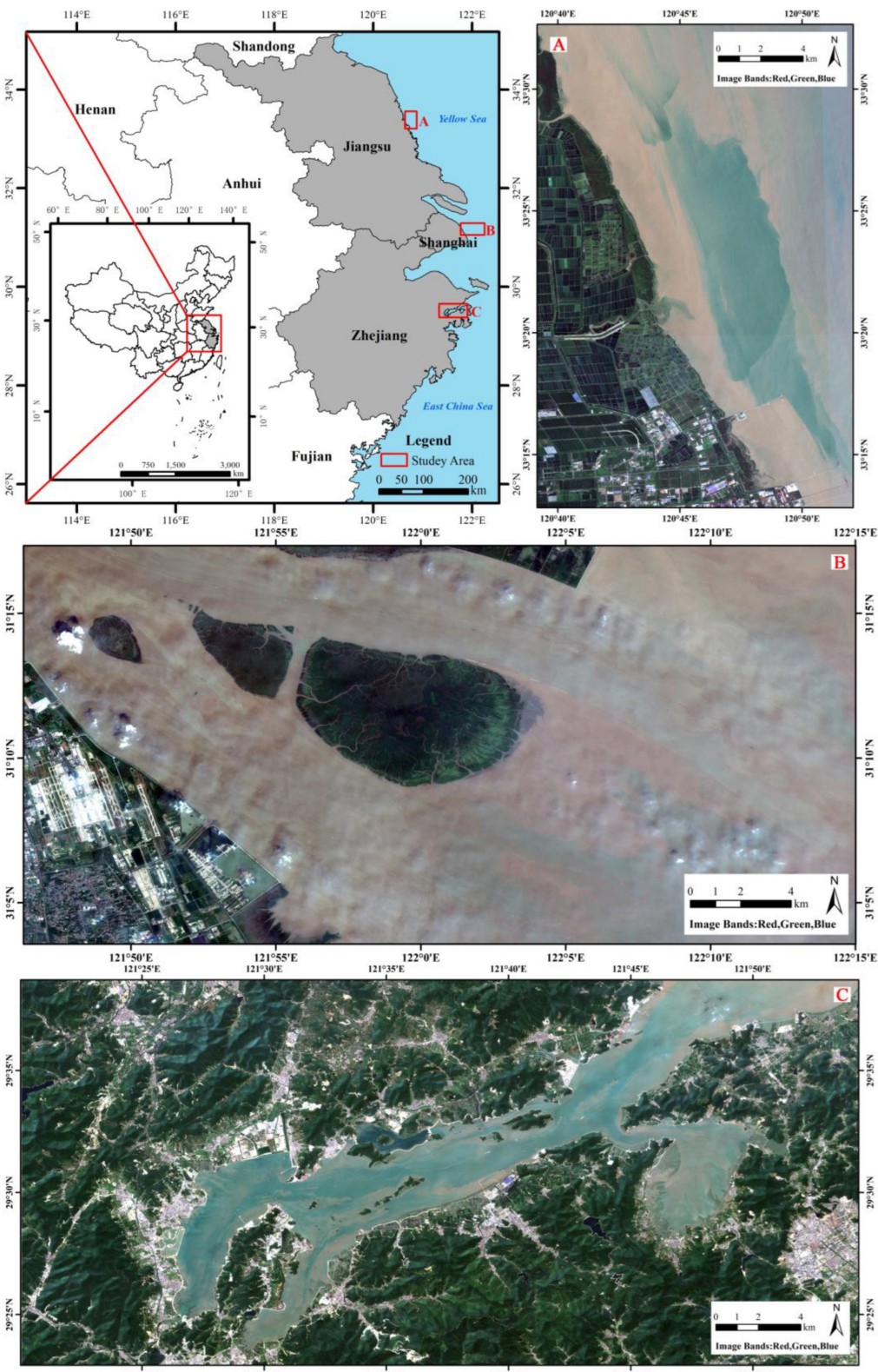

**Figure 1.** Location of the study areas. (**A**) Yancheng plain muddy coast; (**B**) Jiuduansha delta muddy coast; (**C**) Xiangshan harbor muddy coast.

The Yancheng coast is located on the eastern coast of Jiangsu Province, China (120°26′–120°59′E, 33°9′–33°41′N, Figure 1A). The coastal tidal flats are vast, accounting for about 75% of the total tidal flats in Jiangsu Province. The Yancheng coast, which is silt-clayey coast landforms, belongs to the transition zone from the northern subtropical to

the warm temperate climate. There are many fords in the tidal flats. High water content of the tidal flats makes the boundary between tidal flats and seawater not clear. Therefore, it is difficult to extract waterline accurately. The intertidal beach is 500–1000 m wide, where the Spartina alterniflora, Phragmites australis, and Suaeda glauca are widely distributed.

The Jiuduansha coast is located at the estuary of the Yangtze River in China (121°46′–122°15′E, 31°3′–31°17′N, Figure 1B). Currently, the Jiuduansha coast is mainly composed of the Jiangyanansha, Shangsha, Zhongsha, and Xiasha, among which the Zhongsha and Xiasha have merged together, renamed as the Zhongxiasha [42]. These are estuarine sandbars formed by long periods of erosion and deposition due to the interaction of land and ocean. The altitudes of Jiuduansha range from 2.5 m to 3.5 m. Phragmites australis, Spartina alterniflora and Scirpus mariqueter are mainly developed on the beach. Jiuduansha is affected by the sediments carried by the Yangtze River, and the suspended sediment concentration in the nearshore water is high, which also leads to the unclear boundary between tidal flats and water. The channel project and siltation promotion project in Jiuduansha area also changed the original natural evolution of erosion and deposition, making the waterline is in a highly dynamic change.

The Xiangshan coast is located on the eastern coast of Zhejiang Province (121°20′–121°54′E, 29°22′–29°39′N, Figure 1C). It is a long and narrow semi-enclosed bay extending inland which belongs to semi-open harbor muddy coast. Natural muddy coast, erosion coast and artificial coast are distributed alternately in the harbor, which is an important marine aquaculture base with abundant natural resources and marine biological resources. There are a large number of artificial cultivation ponds and rafts in the nearshore area. Due to the spectral similarity between ponds and seawater, the waterline is more likely to be offset to the landside.

### 2.2. Data Source

The Sentinel-2 is a wide-swath imaging mission with high-resolution and multi-spectrum, supporting Copernicus Land Monitoring studies, including the monitoring of vegetation, soil and water cover, as well as observation of inland waterways and coastal areas. The Sentinel-2 Multispectral Instrument (MSI) (Table 1) samples 13 spectral bands: four bands (B2, B3, B4, B8) with a spatial resolution of 10 m, six bands (B5, B6, B7, B8A, B11, B12) of 20 m and three bands (B1, B9, B10) of 60 m. The images covering three study areas had closely acquisition time to minimize the effect of time difference. The overall cloud cover of the images in each study area was less than 5%, and there were no cloud cover in the area at the sea-land interface of the images (Table 1).

**Table 1.** Details of Sentinel-2 MSI data in study areas.

| Data Source | Study Area | Imaging Time (UTC) | Whole Image Cloud Cover | Sea-Land Area Cloud Cover |
|---|---|---|---|---|
| Sentinel-2 MSI | Yancheng | 5 September 2020 02:35:49 | 0.38% | 0% |
| | Jiuduansha | 7 September 2020 02:25:51 | 2.97% | 0% |
| | Xiangshan | 7 September 2020 02:25:51 | 0.79% | 0% |

### 2.3. Methodology

The AEMCW method includes five main steps of pre-processing, band selection, low-frequency information extraction, morphological processing, and post-processing to achieve high-precision and adaptive extraction of complex muddy coast waterline. The flow chart of the AEMCW method is shown as Figure 2.

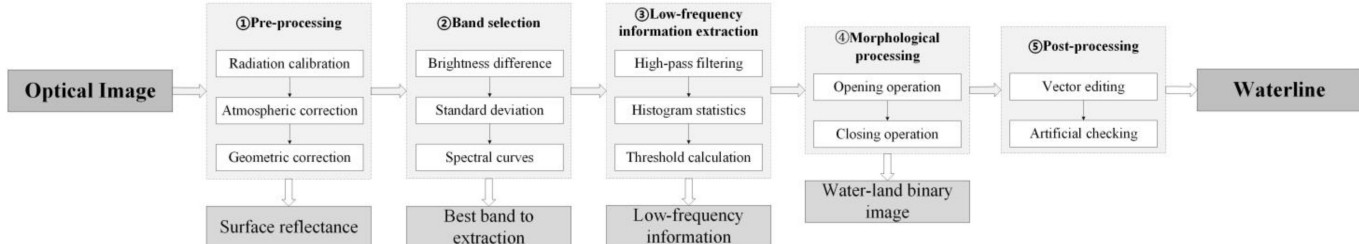

**Figure 2.** Flow chart of the AEMCW (Adaptive Extraction for Muddy Coast Waterline) method.

### 2.3.1. The Pre-Processing of Remote Sensing Image

The pre-processing of remote sensing image includes radiometric calibration, atmospheric correction and orthotopic correction. The Fast Line-of-sight Atmospheric Analysis of Spectral Hypercubes (FLAASH) model parameter was applied to radiometric calibration and atmospheric correction. The orthotopic correction based on Rational Polynomial Coefficient (RPC) was used to obtain surface reflectance data (Figure 1A–C).

### 2.3.2. The Selection of the Best Band

Optical remote sensing image has the characteristics of multiple bands. Different land cover types have different surface reflectance. The greater the difference in surface reflectance between ocean and land, the clearer the boundary between ocean and land. An image with sufficient information is a prerequisite for a large difference in surface reflectance between ocean and land. The greater the brightness difference and standard deviation of the remote sensing image, the more abundant the information contained in the image. In this paper, the minimum, maximum and average values of surface reflectance for each band were first obtained by statistical analysis. The brightness difference (Equation (1)) and standard deviation (Equation (2)) of surface reflectance for each band were further calculated. The corresponding bands with large values of both brightness difference and standard deviation were selected as alternative bands. Then, the spectral curves of water and land in the three study areas were statistically analyzed, and the water bodies were further classified into pond, turbid water and clear water, while the land was further classified into building, non-building and tidal flat. The band with the highest difference in surface reflectance between the two types of features is chosen as the best band for waterline extraction.

$$\text{Brightness Difference} = B_{Max} - B_{Min} \tag{1}$$

$$\text{Standard Deviation} = \sqrt{\frac{1}{N}\sum_{i=1}^{N}\left(x_i - B_{Mean}\right)^2} \tag{2}$$

where $B_{Max}$ and $B_{Min}$ represent the maximum and minimum values of surface reflectance in the band, respectively. N represents the total number of pixels in the band, $x_i$ represents the surface reflectance value of the i-th pixel, and $B_{Mean}$ represents the arithmetic mean of the surface reflectance of the band.

### 2.3.3. Low-Frequency Information Extraction

The remote sensing image has the characteristics of high and low frequency of signals. The edges of the ocean-land boundary correspond to the high-frequency signals of the image, and the intensity of the image varies drastically. The large parts of ocean and land on the image are the areas where the overall intensity changes gently, which correspond to the low-frequency signals of the image. Therefore, the waterline of remote sensing image is the edge information of ocean and land demarcation area. By integrating high-pass filtering, histogram statistics and adaptive threshold determination, the low-frequency information of remote sensing image was obtained. Further combined with morphological

processing, an adaptive remote sensing extraction method of waterline was proposed to achieve high-precision waterline extraction of muddy coast.

(a)    High-pass filtering

Image enhancement techniques for various applications have been rapidly developed in recent years. So far, the common ways to sharpen images are usually by means of the high-pass filter [43]. High-pass filter can increase the difference between low-frequency information and high-frequency information and make the image details more prominent. High-pass filtering (Equation (3)) was applied to the remote sensing image to highlight the high-frequency information and suppress the low-frequency information, thus the difference between the high-frequency and low-frequency information were enlarged to highlight the waterline of the image. Specifically, the center element of the high-pass filter convolution kernel was set to 8, all the surrounding elements were set to −1, and the sum of all elements in the convolution kernel was 0.

$$H = \begin{bmatrix} -1 & -1 & -1 \\ -1 & 8 & -1 \\ -1 & -1 & -1 \end{bmatrix} * I \tag{3}$$

where H represents the result of high-pass filtering and I denotes the original remote sensing image surface reflectance data.

(b) Histogram statistics

Histogram statistics were performed on the pixel values of image H to obtain the distribution of high and low frequency information (Figure 3). The single-band remote sensing image is a gray-scale image, which is generally divided into 256 gray levels from black to white with 255 intervals between the maximum value and the minimum value. Therefore, we divided the histogram of the single-band remote sensing image into 255 pixel-intervals. The group distance of the histogram can be calculated as Equation (4).

$$S = \frac{H_{Max} - H_{Min}}{255} \tag{4}$$

where $H_{Max}$ and $H_{Min}$ represent the maximum and minimum values of image H pixels, respectively.

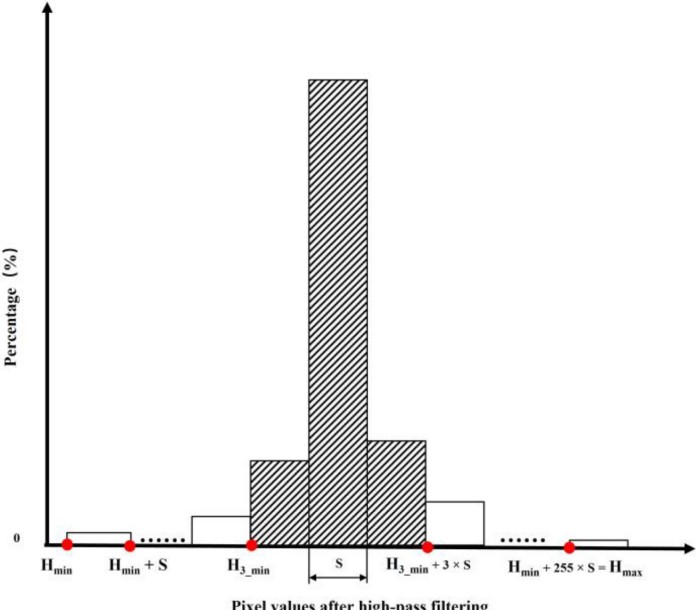

**Figure 3.** Histogram of high-pass filtering result.

(c) Adaptive threshold determination

Adaptive threshold determination can help to make the threshold more accurate [33,34]. After high-pass filtering, the histogram of the image in the intersection area of sea and land presented extremely significant single-peak characteristics. The single-peak represents the low-frequency information in the image. According to the the distribution of the histogram (Figure 3), the top three pixel-intervals with the highest percentage can approximately represent the single-peak. Therefore, the threshold range of single-peak can be determined adaptively with Equation (5). The smallest pixel value of the top three pixel-intervals with the highest percentage was taken as the left-hand point, noted as $H_{3\_min}$, thus the low-frequency information threshold range can be expressed as $[H_{3\_min}, H_{3\_min} + 3 \times S]$.

$$(H \text{ ge } H_{3\_min}) \text{ and } (H \text{ le } (H_{3\_min} + 3 \times S)) \tag{5}$$

where H represents the high-pass filtered pixel value, S represents the group distance of the histogram, ge denotes greater than or equal, and le denotes less than or equal.

2.3.4. Morphological Processing

There were many isolated dots and tiny holes in low-frequency information images. Isolated dots were set with an area $\leq 10$ pixels in size caused by low-frequency information, and tiny holes were set with an area $\leq 10$ pixels caused by high-frequency information. Morphological filtering including opening and closing operators was applied to the binary image after low-frequency information extraction to obtain complete binary image results of water and land separation [44–46]. Specifically, the morphological opening operation (i.e., erosion operation (Equation (6)) followed by dilation operation (Equation (7)) to remove isolated dots, and the morphological closing operation (i.e., dilation operation followed by erosion operation) to fill the tiny holes.

$$C \ominus D = \{(m,n) | (D)_{mn} \subseteq C\} \tag{6}$$

$$C \oplus D = \{(m,n) | (D)_{mn} \cap C \neq \varnothing\} \tag{7}$$

where $\ominus$ denotes erosion operation symbol, and $\oplus$ denotes dilation operation symbol. C represents the extraction result of low-frequency information. D represents the convolution kernel of morphological operation. The size of the convolution kernel is $3 \times 3$, and the value of each element is 1. m represents the horizontal position of the pixel in the image. n represents the vertical position of the pixel in the image. $(D)_{mn}$ represents the position of the convolution kernel D in the image when the core center of the convolution kernel D is located at (m,n). In the extraction result of low-frequency information, the value of low-frequency information is 1, and the value of high-frequency information is 0.

2.3.5. Post-Processing

Morphological processing can remove most of the isolated dots and fill a majority of the tiny holes in low-frequency information images, but some isolated dots and tiny holes cannot be completely removed by a single morphological operation due to their large size. Additionally, multiple morphological operations tend to lose more detailed boundary information. Therefore, after the morphological processing, we converted the binary image of the water-land separation into a vector format for post-processing. First, polygons were selected according to their area and properties (Equation (8)), in order to eliminate surfaces with smaller areas and obtain the final waterline extraction results,

$$(\text{area} \geq 0.05) \text{ and } (\text{GRIDCODE} = 1) \tag{8}$$

where area represents the polygon area and is set to an area of 0.05 Decimal Degrees. This is an empirical threshold that can be adjusted. GRIDCODE indicates the polygon attribute, and it takes a value of 1 for water and 0 for land.

2.3.6. Accuracy Assessment

Accuracy assessment includes length accuracy assessment and spatial accuracy assessment. The waterline obtained from VI was selected as the reference waterline. The length error $\Delta_L$ between the extracted waterline and the reference waterline was calculated for length accuracy assessment with Equation (9). Creating a buffer zone based on the reference waterline is the first step to assess spatial accuracy. The buffer radius was set to 1 pixel width of the corresponding image. The qualified extraction waterline is defined as the part where the extraction waterline intersects with the buffer zone. The indicators such as producer accuracy (PA, Equation (10)), user accuracy (UA, Equation (11)) and $F_1$ score (Equation (12)) were used to assess the spatial accuracy. $F_1$ score is an overall metric that combines producer and user accuracy, which indicates more accurate extraction results with larger values [47].

$$\Delta_L = \frac{E - T}{T} \tag{9}$$

$$PA = \frac{T \cap E}{T} \tag{10}$$

$$UA = \frac{T \cap E}{E} \tag{11}$$

$$F_1 \text{ score} = 2 \times \frac{PA \times UA}{PA + UA} \tag{12}$$

where T is the length of the reference waterline obtained by VI, and E is the length of the actual extracted waterline. $T \cap E$ denotes the total length of the portion of the extracted waterline that intersects with the buffer zone.

## 3. Results

### 3.1. The Best Band to Extract Waterline

The spectral information of single-band from Sentinel-2 MSI images for the three study areas of Yancheng, Jiuduansha and Xiangshan coast are shown in Table 2, and the spectral curves for the six categories of pond, turbid water, clear water, building, non-building and tidal flat for the three study areas are shown in Figure 4.

**Table 2.** Details of Sentinel-2 MSI data in study areas.

| Band | Yancheng | | Jiuduansha | | Xiangshan | |
| | Brightness Difference | Standard Deviation | Brightness Difference | Standard Deviation | Brightness Difference | Standard Deviation |
|---|---|---|---|---|---|---|
| B1 | 0.3411 | 0.024365 | 0.7222 | 0.021560 | 0.3715 | 0.023341 |
| B2 | 0.7320 | 0.031482 | 0.7711 | 0.024222 | 0.9221 | 0.030383 |
| B3 | 0.8312 | 0.036366 | 0.7526 | 0.025450 | 0.9975 | 0.032860 |
| B4 | 0.9344 | 0.053356 | 0.9509 | 0.034976 | 0.9955 | 0.041890 |
| B5 | 0.9008 | 0.043899 | 0.8041 | 0.026310 | 0.9799 | 0.038235 |
| B6 | **0.9292** | **0.079393** | **0.8024** | **0.056261** | 0.9752 | 0.090276 |
| B7 | 0.8303 | 0.101010 | 0.7351 | 0.067939 | **0.9849** | **0.113368** |
| B8 | **0.9889** | **0.107864** | 0.7416 | 0.074885 | **0.9803** | **0.119171** |
| B8A | 0.8709 | 0.112967 | 0.7582 | 0.084939 | **0.9950** | **0.128550** |
| B9 | 0.6437 | 0.116672 | **0.9865** | **0.100055** | 0.5817 | 0.128314 |
| B11 | **0.9748** | **0.071684** | **0.9644** | **0.073782** | **0.9960** | **0.081913** |
| B12 | 0.9667 | 0.049392 | 0.9526 | 0.054115 | 0.9971 | 0.058792 |

The bands with brightness difference and standard deviation in the top 50% were firstly selected as the alternate bands. For Yancheng coast, the bands were B6, B8 and B11. For Jiuduansha coast, the bands were B6, B9 and B11, for the Xiangshan coast, the bands were B7, B8A, B8 and B11. Considering the spectral curves of the land cover types, B6, B7, B8 and B8A bands have similar features as tidal flat, turbid water and clearwater in the

Yancheng coast and the Jiuduansha coast, while B9 and B11 have larger surface reflectance differences between the land and water in the three study areas, which can better highlight the location of the waterline. In addition, the spatial resolution of B9 is 60 m, far lower than the 20 m of B11. Conversely, the B11 with a central wavelength of 1610.40 nm was selected as the best band for waterline extraction. Previous studies have also shown that near-infrared bands easily confuse waterline with whitewater, while short-wave infrared bands are more reliable [48].

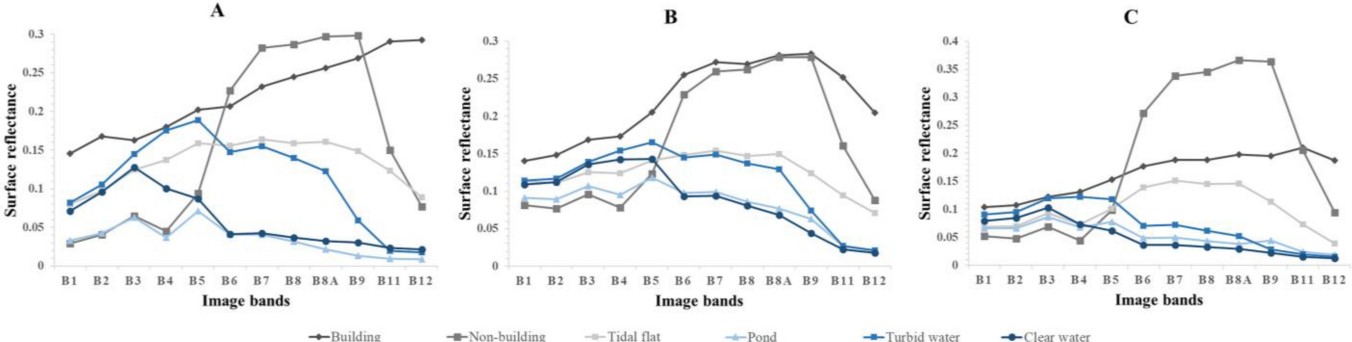

**Figure 4.** Spectral curves of land cover types in the three study areas. (**A**) Yancheng plain muddy coast; (**B**) Jiuduansha delta muddy coast; (**C**) Xiangshan harbor muddy coast.

### 3.2. Waterline Extraction Results of Muddy Coast

The B11 after high-pass filtering in the Yancheng coast images are shown in Figure 5A. Figure 6 is the corresponding histogram. The $H_{Max}$ and $H_{Min}$ were 6.16 and −3.77, respectively. The S of the histogram was 0.0389. The $H_{3\_min}$ was −0.0704, thus the threshold range of low-frequency information was [−0.0704, 0.0464]. The results of low-frequency information extraction are shown in Figure 5B. The binary image of water and land separation obtained after further morphological processing is shown in Figure 5C.

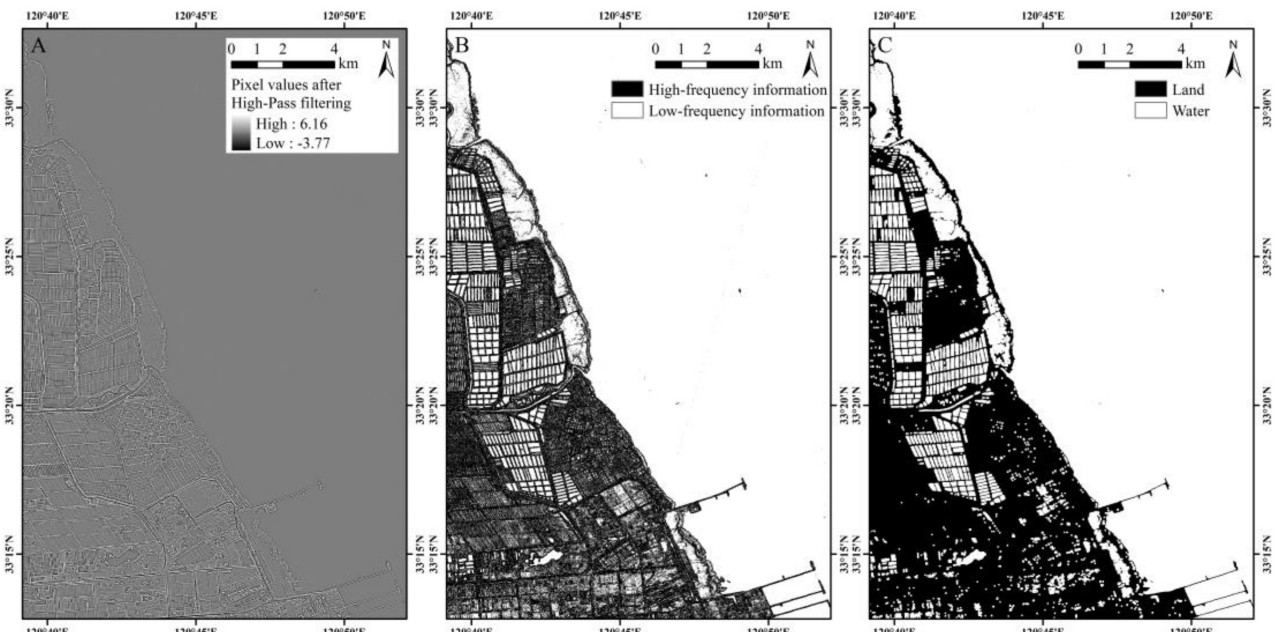

**Figure 5.** Extraction process of Yancheng Coast. (**A**) High-pass filtering result; (**B**) Low-frequency information; (**C**) Water-land binary image.

The waterline results of Yancheng Coast, Jiuduansha Coast and Xiangshan Coast extracted by the proposed method are shown as Figure 7A–C, respectively.

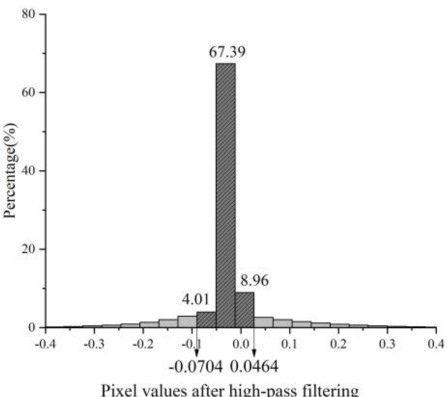

**Figure 6.** The histogram of the high-pass filtered image.

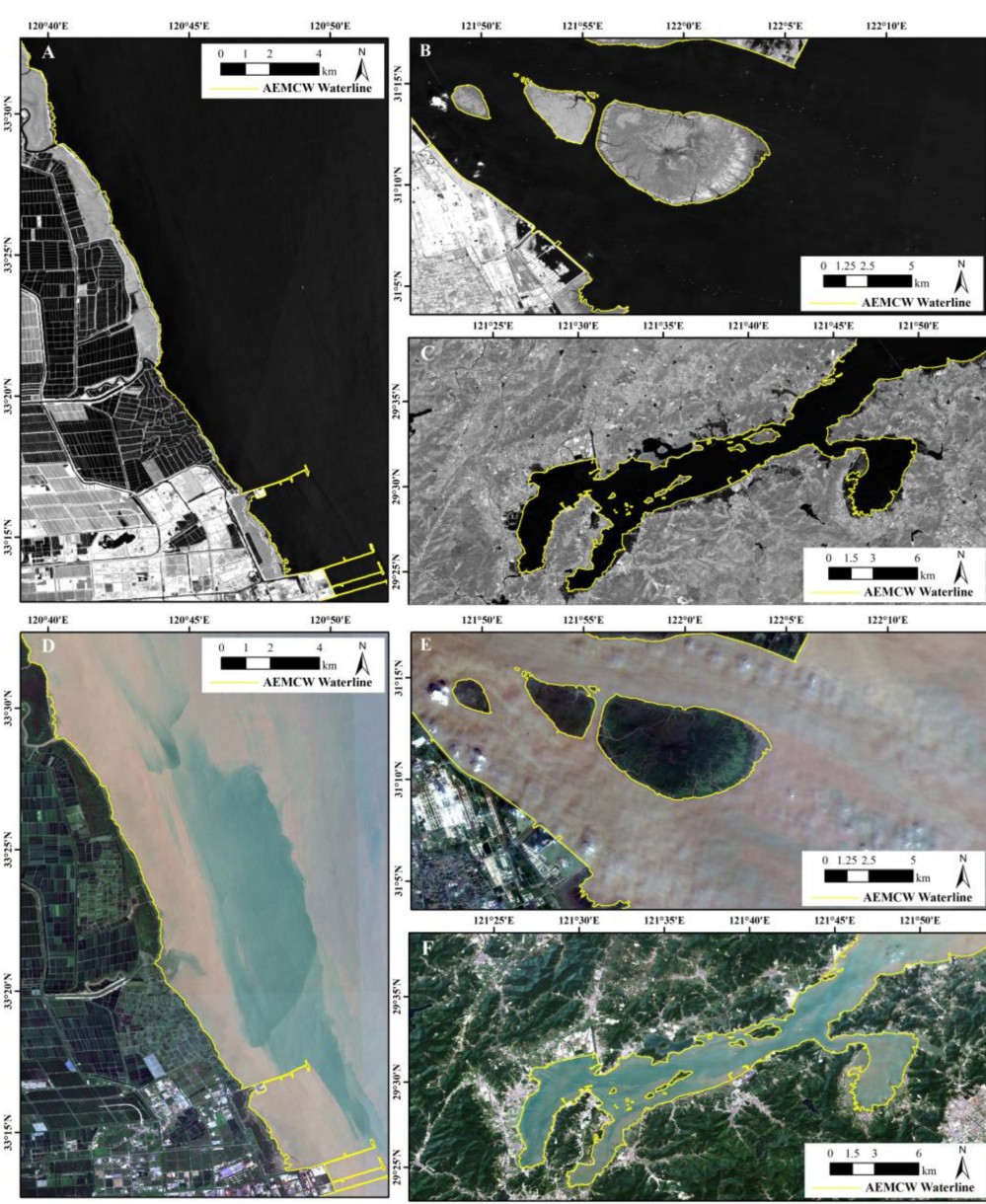

**Figure 7.** Waterline results of the AEMCW method. (**A**,**D**) Yancheng plain muddy coast; (**B**,**E**) Jiudu-ansha delta muddy coast; (**C**,**F**) Xiangshan harbor muddy coast; (**A–C**) are the single-band images with the band B11. (**D–F**) are the multiband images with the band combination of band B4, B3 and B2.

## 4. Discussion

### 4.1. Accuracy Assessment

4.1.1. Threshold Settings and Extraction Results

The waterlines extracted by NDWI, MNDWI and ED were applied for comparison and validation with the waterlines extracted by AEMCW method. The optimal water-land separation thresholds for NDWI and MNDWI were determined by iterative adjustments. NDWI thresholds were −0.08, −0.1 and 0.15, MNDWI were 0.57, 0.58 and 0.58 in the three study areas, respectively. With regards to the ED for waterline extraction, the canny operator that is not susceptible to noise interference and can detect true weak edges was chosen [49], and the threshold range was set as [0.064, 0.16].

The results of waterline extraction by each method in the three coasts are shown in Figure 8A–C, respectively. Regarding coasts with clear water-land boundaries, the spatial locations of the waterlines were generally consistent (Figure 8(A2,B2,C1,C4)), but the waterline results extracted from NDWI and MNDWI had large deviations on local coasts. The NDWI method did not extract the piers on the Yancheng Coast (Figure 8(A3)). Some coastal shores were identified as water bodies due to the small difference in NDWI values (Figure 8 NDWI), causing the waterline to be shifted towards the landward (Figure 8(B3)). The NDWI and MNDWI methods also identified some of the nearshore water bodies as land, causing the waterline to be shifted towards the seaward (Figure 8(A1,B1)). The waterline extracted by the AEMCW had a small deviation from the reference waterline in some complex coastal areas where farming ponds and tidal flats distributed (Figure 8(C3)).

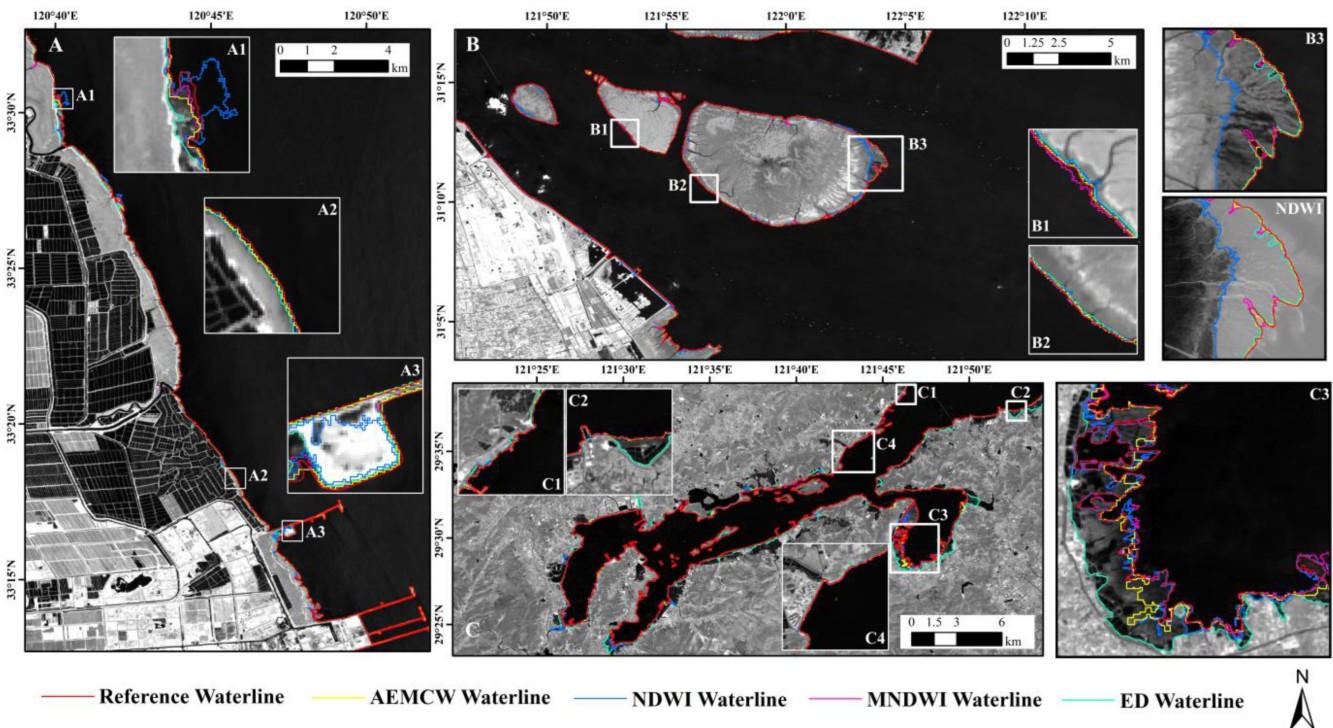

**Figure 8.** Waterline results of the AEMCW, NDWI (Normalized Difference Water Index), MNDWI (Modified Normalized Difference Water Index) and ED (Edge Detection) methods. (**A**) Yancheng plain muddy coast; (**B**) Jiuduansha delta muddy coast; (**C**) Xiangshan harbor muddy coast; (**A1**–**A3**,**B1**–**B3**,**C1**–**C4**) are the enlarged view.

4.1.2. Length Accuracy Analysis

The length errors of the waterlines in all three study areas were 14.4%, 18.0% and 7.7% for the AEMCW method, respectively, all below 20%, while the length errors of the NDWI and MNDWI index methods were almost above 30%. The ED method achieved the

smallest length error of the waterlines (Table 3). Due to the waterlines extracted by these methods based on pixel information, the waterlines show a distinctly serrated character. Conversely, the waterline obtained by VI method was smoother. Therefore, for the same section of waterline, there was a relatively big difference between the automatic extraction algorithm and the VI method in the length of the waterline.

**Table 3.** Length accuracy of the AEMCW and traditional methods.

| Method | Yancheng | | Jiuduansha | | Xiangshan | |
|---|---|---|---|---|---|---|
| | Length (km) | Error (%) | Length (km) | Error (%) | Length (km) | Error (%) |
| VI | 92.781 | - | 145.643 | - | 305.648 | - |
| AEMCW | 106.126 | 14.4% | 171.810 | 18.0% | 329.175 | 7.7% |
| NDWI | 97.338 | 4.9% | 192.908 | 32.5% | 463.655 | 51.7% |
| MNDWI | 121.770 | 31.2% | 199.032 | 36.7% | 430.566 | 40.9% |
| ED | **92.566** | **−0.2%** | **147.815** | **1.5%** | **284.900** | **−6.8%** |

4.1.3. Spatial Accuracy Analysis

Regarding the spatial accuracy analysis, the producer accuracies of AEMCW method in the three study areas were 94.3%, 109.6% and 94.2%, respectively. The user accuracies were 82.4%, 92.9% and 87.5%, respectively. The $F_1$ score values were 88.0%, 100.6% and 90.7%, respectively (Table 4). Among them, the waterline producer accuracy of the Jiuduansha coast exceeds 100%. This is because the qualified waterline is defined as the intersect part of the extracted waterline with the buffer of the reference waterline. The qualified waterline does not completely overlap the reference waterline, resulting in the length difference. The accumulation of the length differences makes the total length of the qualified waterlines greater than that of the reference waterlines, causing the producer accuracy greater than 100%.

**Table 4.** Spatial accuracy of the AEMCW and traditional methods.

| Study Area | Method | PA | UA | $F_1$ Score |
|---|---|---|---|---|
| Yancheng | AEMCW | 94.3% | **82.4%** | **88.0%** |
| | NDWI | 57.6% | 54.9% | 56.2% |
| | MNDWI | **101.2%** | 77.1% | 87.5% |
| | ED | 47.4% | 47.5% | 47.5% |
| Jiuduansha | AEMCW | 109.6% | **92.9%** | **100.6%** |
| | NDWI | 55.6% | 42.0% | 47.9% |
| | MNDWI | **118.3%** | 86.6% | 100.0% |
| | ED | 37.9% | 37.3% | 37.6% |
| Xiangshan | AEMCW | 94.2% | **87.5%** | 90.7% |
| | NDWI | 102.7% | 67.7% | 81.6% |
| | MNDWI | **111.9%** | 79.5% | **92.9%** |
| | ED | 35.9% | 38.5% | 37.1% |

The $F_1$ score values of the NDWI method were less than 60% for both the Yancheng coast and the Jiuduansha coast (Table 4), which were much lower than those of the AEMCW method (greater than 85%). According to the spectral curves of land cover types (Figure 4B–D), the Yancheng coast and the Jiuduansha coast were both greatly affected by the riverine sediments. Thus, the turbid water and tidal flats have similar surface reflectance values at B3 (560 nm) and B8 (842 nm) band. This similar spectral feature results in the smaller NDWI difference between turbid water and tidal flats, causing poor adaptability of NDWI for the complex muddy coasts. On the Xiangshan coast, due to the influence of the island's barrier, the seawater inside the harbor is clear. The distinction between turbid water and tidal flats is greater at B8 band, the difference between land and water is more obvious. Therefore, the accuracy of waterline extraction is significantly improved. $F_1$ score of NDWI method reached 81.61%. The previous research on muddy coast

also shows that the spectral characteristics of tidal flats and water bodies in the 545~565 nm band are highly similar, while the short-wave infrared band has the best differentiation among the tidal flat water, soil, and vegetation. The mapping accuracy of NDWI for muddy coastline is low [6]. In addition, the use of the global NDWI threshold can ensure the accuracy of the water-land separation in a large space region, while in a local area, it cannot achieve accurate water-land separation.

The bands for the MNDWI calculation were B3 and B11 (1610 nm), which have achieved the highest producer accuracy in the three study areas (Table 4). The spatial accuracies of MNDWI ($F_1$ score of 87.5%) and AEMCW ($F_1$ score of 88.0%) were comparable (Table 4). However, the length error of MNDWI method was much larger than that of AEMCW method, which indicated that the waterline extracted by AEMCW method has less redundant information and less data volume than that extracted by MNDWI when the spatial accuracies of the two methods are similar. Regarding the Yancheng coast and the Xiangshan coast, the user accuracies of MNDWI were 77.1% and 79.5%, both were less than 80%. Compared with NDWI, the MNDWI has achieved better extraction accuracy. This is because the MNDWI is only affected by the spectral similarity of band B3 to turbid water and tidal flats, while the NDWI is affected by both B3 and B8 discussed in the previous paragraph. In addition, the MNDWI is also affected by the global threshold, resulting in the low extraction accuracy of the waterlines. Unlike the NDWI and MNDWI, the AEMCW used B11 to extract waterline. This short-wave infrared band can distinguish the turbid water and tidal flats in spectral features. Therefore, the AEMCW method achieves better spatial accuracy than the NDWI and MNDWI methods. The introduction of low-frequency information threshold extraction formula enables AEMCW method to adaptively obtain more accurate thresholds according to the spectral characteristics of the images themselves, and reduces the influence of local inaccuracies caused by using global thresholds.

The producer accuracy and user accuracy of the ED method in the three study areas were both lower than 50% (Table 4), and the waterline offsets more to the land side compared to the reference waterline (Figure 8(A1,C2,C3)). Take Figure 8(C3) as an example, there are ponds and tidal flats at the junction of land and water. In the B11, the surface reflectance of the ponds (0.024) and tidal flats (0.073) are both between the clean water (0.015) and buildings (0.210). The turbid water (0.019) is similar to clean water. In the complex muddy coast, the actual waterline is between the tidal flats and the turbid water. Compared with the difference of spectral characteristics between ponds or tidal flats and buildings, the difference between tidal flats and turbid water is smaller, resulting in stronger edge information between ponds or tidal flats and buildings. Therefore, the location of waterline extracted by edge-detection algorithm offsets more to the land side. To verify the above deduction, we further expanded the buffer radius to 2 pixels for spatial accuracy assessment (Table 5), of which one contained the left and right sides of the reference waterline, and the other only contained the land side. Compared with the double-side buffer with a buffer radius of 1 pixel, the $F_1$ score values of that with 2 pixels on the three study areas increased by 32.4%, 49.6% and 22.4%, respectively. The $F_1$ score values of single-side buffer with 2 pixels increased by 19.3%, 46.0%, and 16.8%, respectively. The parts of the increased accuracy were more from the land single-side buffer, which indicate that the waterline extracted by the ED offsets to the land side. The waterline extraction process of the AEMCW is different with the ED. By high-pass filtering, the sea surface was processed as low-frequency information. The AEMCW method obtained the waterline by morphological and post processing of the low-frequency information region, morphological and post processing can remove tiny polygons, which can effectively reduce the influence of tiny polygons' edges on the determination of waterline position. Therefore, the AEMCW method can effectively avoid the offset of the extracted waterline to the land side. Moreover, the waterline obtained by the ED method is extremely discontinuous, which requires more post-processing. In comparison, the AEMCW method replaces the direct extraction of edge information by extracting the boundary of low-frequency information, so that the waterline is continuous. The AEMCW method still has some drawbacks. In the

image of land and sea interface area, large area of seawater is low-frequency information, after high-pass filtering, the single-peak representing low-frequency information is very prominent. However, in some images with less low-frequency information, such as inland river regions, the low-frequency information is relatively less, the single-peak will become gentle. This may cause AEMCW can not achieve a good extraction effect.

**Table 5.** Spatial accuracy of the ED method with different buffers.

| Study Area | | PA | Δ | UA | Δ | $F_1$ Score | Δ |
|---|---|---|---|---|---|---|---|
| **Yancheng** | 1 pixel Buffer | 47.4% | - | 47.5% | - | 47.5% | - |
| | 2 pixels Buffer | 79.8% | 32.4% | 80.0% | 32.5% | 79.9% | 32.4% |
| | 2 pixels Land Buffer | 66.7% | 19.3% | 66.8% | 19.3% | 66.7% | 19.3% |
| **Jiuduansha** | 1 pixel Buffer | 37.9% | - | 37.3% | - | 37.6% | - |
| | 2 pixels Buffer | 87.9% | 50.0% | 86.6% | 49.3% | 87.2% | 49.6% |
| | 2 pixels Land Buffer | 84.2% | 46.3% | 82.9% | 45.6% | 83.5% | 46.0% |
| **Xiangshan** | 1 pixel Buffer | 36.4% | - | 41.3% | - | 38.7% | - |
| | 2 pixels Buffer | 57.5% | 21.1% | 65.2% | 23.9% | 61.1% | 22.4% |
| | 2 pixels Land Buffer | 52.2% | 15.8% | 59.2% | 17.9% | 55.5% | 16.8% |

Δ denotes the difference between the accuracy index and the corresponding double-side buffer accuracy index with a buffer radius of 1 pixel.

### 4.2. Application of Proposed Method in Hyperspectral Image

In order to verify the effectiveness and robustness of the proposed method, ZY-1 02D hyperspectral satellite Advanced Hyperspectral Imager (AHSI) images, acquired at 02:56:07, 6 September 2020, UTC time in the Yancheng coast were applied. There was no cloud coverage in the land-water junction area. According to the best band (center wavelength 1610.40 nm) determined in the Sentinel-2 MSI image, the optimal band of the ZY-1 02D was determined to be the 37th band with center wavelength of 1610.39 nm. The same preprocessing algorithm as the Sentinel-2 MSI image was used to the ZY-1 02D AHSI image. Geometric correction was based on the Sentinel-2 MSI image with higher spatial resolution, and the tie points were generated through the cross-correlation matching algorithm. The image geometry model was the Fitting Global Transform, and the minimum matching point was set to 0.6, the maximum allowable error of the connection point was 5. The ZY-1 02D image was geometrically registered through a First-Order Polynomial transformation model, and the registration accuracy was not exceeding 1 pixel (Figure 9A). Using the AEMCW method, we obtained the waterline of the ZY-1 02D AHSI image on the Yancheng Coast (Figure 9B).

According to the accuracy assessment, the length error of waterline extraction result was 9.9%. The producer accuracy and the $F_1$ score were 89.8% and 85.6%, respectively, which were all higher than 85%. It shows that the proposed method could achieve a high-precision waterline extraction result both for multispectral and hyperspectral optical remote sensing images, even the image with different resolutions. Therefore, the proposed algorithm for muddy waterline extraction is robust and transferable.

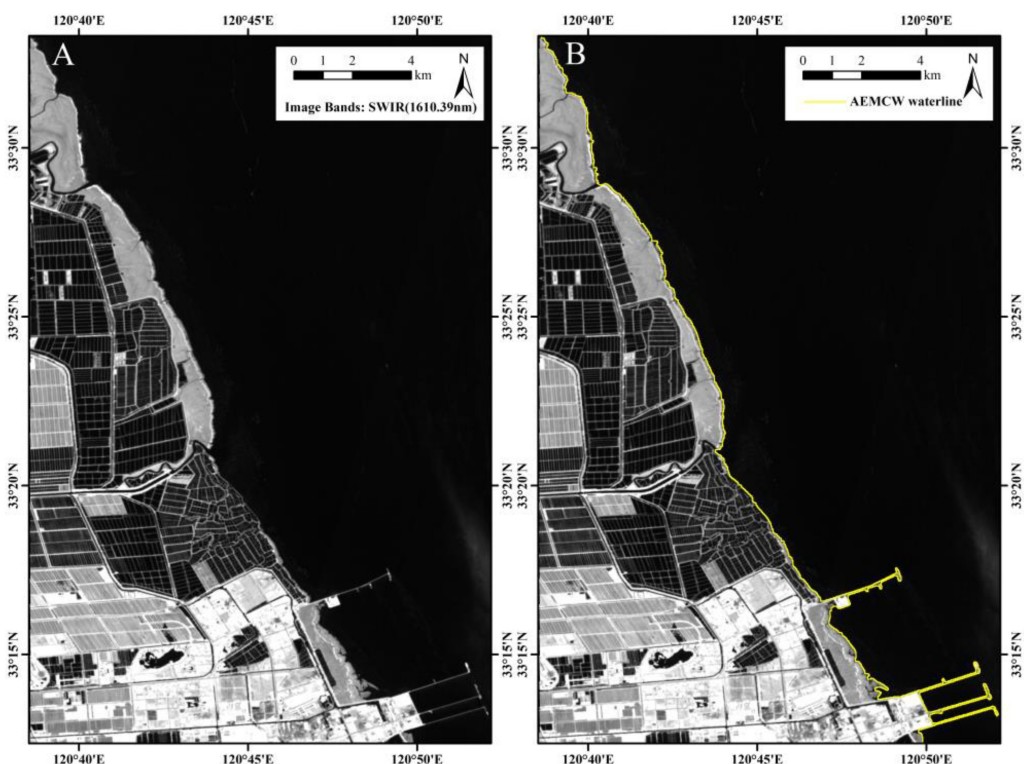

**Figure 9.** ZY-1 02D image and waterline results. (**A**) The SWIR (1610.39nm) band of ZY-1 02D; (**B**) The waterline extracted by AEMCW method of ZY-1 02D in Yancheng plain muddy coast.

## 5. Conclusions

The waterline extraction in the complex muddy coast is affected by the dual effects of high turbidity water in the sea and the high water content of the tidal flats on the land, making the extraction of the muddy coast waterline with high dynamic changes challenging. Therefore, the paper proposed an adaptive remote sensing extraction named AEMCW, which mainly involves high-pass filtering, histogram statistics, adaptive threshold determination and morphological processing. The three muddy coasts of different landform in the Yancheng Coast, the Jiuduansha Coast and the Xiangshan Coast were selected as the study areas. The results showed that the length errors of AEMCW in the three study areas were 14.4%, 18.0% and 7.7%, respectively. The producer accuracies were 94.3%, 109.6%, and 94.2%, respectively. The user accuracies were 82.4%, 92.9%, and 87.5%, respectively. And the $F_1$ score values were 88.0%, 100.6%, and 90.7%, respectively. Comparison with NDWI, MNDWI and ED methods indicated that the AEMCW algorithm could achieve the low length error, high spatial accuracy and has the unique advantage to obtain waterlines of muddy coast. It can determine the low-frequency information extraction threshold adaptively according to the spectral characteristics of the image itself without the repeatedly adjustment the threshold of different images. In addition, the method can obtain the edge information by extracting the low-frequency information boundary, and can ensure the continuity of the waterline results. Moreover, the waterline results have the advantages of low redundant information and small data volume for different types of muddy coasts. Furthermore, application of this method to hyperspectral remote sensing image demonstrated that AEMCW method has good robustness. In the future, we plan to edit code in remote sensing cloud platform to implement AEMCW method and apply it to the long-time series study of the evolution of erosion and deposition.

**Author Contributions:** Conceptualization, L.W. and W.S.; Methodology, Z.Y., L.W. and W.S.; Software, Z.Y.; Validation, W.X.; Formal Analysis, L.W.; Investigation, G.Y. and C.C.; Resources, G.Y. and C.C.; Data Curation, Z.Y.; Writing—Original Draft Preparation, Z.Y.; Writing—Review and Editing, L.W., W.S., W.X., B.T., Y.Z., G.Y. and C.C.; Visualization, Z.Y.; Supervision, G.Y. and C.C.; Project Administration, L.W. and W.S.; Funding Acquisition, L.W., B.T. and Y.Z. All authors have read and agreed to the published version of the manuscript.

**Funding:** This research was funded by the Open Research Fund of State Key Laboratory of Estuarine and Coastal Research (No. SKLEC-KF202104), in part by the National Natural Science Foundation of China (Nos. 42176174, 42122009, 41971296, 42171311), in part by the Zhejiang Provincial Natural Science Foundation of China (Nos. LY22D010002, LR19D010001), in part by the National Science Foundation for Post-doctoral Scientists of China (No. 2020M683258) and in part by the Chongqing Technology Innovation and Application Development Special Project (No. cstc2020jscx-msxmX0193).

**Institutional Review Board Statement:** Not applicable.

**Informed Consent Statement:** Not applicable.

**Data Availability Statement:** All Sentinel-2 Surface Reflectance data used in this analysis can be accessed with the Google Earth Engine (https://developers.google.com/earth-engine/datasets/catalog/COPERNICUS_S2_SR, accessed on 1 January 2022). All ZY-1 02D data used in this analysis is not applicable.

**Acknowledgments:** The authors sincerely thank all anonymous reviewers who provided detailed and valuable comments or suggestions to improve this manuscript.

**Conflicts of Interest:** The authors declare no conflict of interest.

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
