# Peer review of "A New Adaptive Remote Sensing Extraction Algorithm for Complex Muddy Coast Waterline"

_remotesensing, doi:10.3390/rs14040861_

Round 1

Reviewer 1 Report

The research purpose is well stated in this paper and it is well written. The text is well-structured and the presentation is referenced. The study areas present test cases with different characteristics and so the robustness of the proposed algorithm is explored in the right direction.

Some comments:

- Authors state that the proposed algorithm could achieve the adaptive extraction of the muddy coast waterline. How is this statement established? In what way the proposed steps of the algorithm guarantee accurate adaptive extraction of the muddy coast waterline? For example, there was no study presented in the state of the art (Introduction section) that used adaptive threshold determination or any adaptive determination of parameters? In other words, the contribution of this paper should be clearer in the Introduction section. Why the presented work is unique from existing works, what is the novelty?

- The ‘2.3 Section’ should be referenced, at least at the basic ‘state of the art’ methods that are used (e.g. High-pass filtering and how the filter is convolved to the image, adaptive thresholding, opening operation, etc).

- ‘2.3.2 Section’: The spectral curves of water were statistically analyzed for what reason? In order to detect water bodies? Did the authors use any reference spectral curves of water? If so, please mention and reference. Also, how was the classification of the water bodies into pond, turbid water and clear water performed? Using human (manual) perception or using any automated way?

- ‘2.3.4 Section’: What is the shape of the kernel used for the opening operation?

- ‘2.3.5 Section’: What are the parameters that the empirical threshold depends on? How was it selected for the presented study? Why is it appropriate for all three study areas?

- ‘3. Section’: It would be helpful if there was an extra figure under Figure 7 showing the waterline results located over the original images that are presented in Figure 1. In this way, the muddy coast with the resulted waterline would be better presented to the reader.

- ‘4.1.1. Threshold settings and extraction results Section’: What color is the AEMCW waterline indicator? Unfortunately, even with large page zoom it is not easily visible at some areas of Figure 8. Please try to change the specific color and correct, if possible, the figure’s image analysis.

- Do the authors have any plans of future work, since the proposed study looks promising?

Reviewer 2 Report

This paper proposed an adaptive remote sensing extraction algorithm framework for the complex muddy coast waterline, which was validated in three case areas. The paper is well organized, the language is relatively fluent, and a considerable amount of comparative experiments are performed, a very impressive workload. However, the paper still needs to be responded and revised for the following issues.

  1. The Abstract background knowledge is too lengthy and it is suggested to further compress it into 1-2 sentences. The authors conducted a considerable amount of comparative experiments with multiple methods and I suggest summarizing it in the Abstract.
  2.  F1 score should be expressed in another way here, it is easy to be confused before it is defined. The last sentence of the application prospect needs further revision.
  3. The last paragraph of the Introduction needs to clarify the innovation of the method and the contribution of the paper.
  4. Study area, the paper mentioned that the difficulty of shoreline extraction is the influence of natural conditions and human activities. Could this be described in three case areas, which might be more interesting for the readers.
  5. the subheadings in the 2.3 Methods section are confusing. the next level of headings in the 2.3.3 section are 1,2,3? two repetitions of 2.3.5? the authors need to check and revise.
  6. In the Discussion section, it is suggested that the authors should explain clearly the reasons for the better results of the proposed method compared to other methods, and it is also hoped that the authors can identify the limitations of the proposed method and the areas that need further improvement in the future.
